# Utilizing random forest algorithm for early detection of academic underperformance in open learning environments

Shikah Abdullah Albriki Balabied[1] and Hala F. Eid[2]

[1] Department of Quality of Life and Continuing Education, College of Education and Human Development, University of Bisha, Bisha, Saudi Arabia

[2] Department of Business Administration, Applied College, University of Bisha, Bisha, Saudi Arabia

## ABSTRACT

**Background:** One of the primary benefits of Open Learning Environments (OLEs) is their scalability. OLEs provide flexible and accessible learning opportunities to a large number of students, often on a global scale. This scalability has led to the development of OLEs that cover a wide range of subjects and disciplines, from computer science and engineering to humanities and social sciences. However, the scalability of OLEs also presents some challenges *i.e.*, it can be too difficult to provide personalized support and feedback to individuals. Early prediction of student performance can improve the learning experience of students by providing early interventions and support.

**Method:** The specific objective of this study was to build a model that identifies at-risk students and allows for timely interventions to promote their academic achievement. The random forest classifier model has been used for analyzing anonymized large datasets available from Open University Learning Analytics (OULAD) to identify patterns and relationships among various factors that contribute to student success or failure.

**Results:** The findings of this study suggest that this algorithm achieved 90% accuracy in identifying students who may be at risk and providing them with the necessary support to succeed.

## INTRODUCTION

The last decade has seen a significant amount of Open Learning Environments (OLEs) development *e.g.*, Massive Open Online Courses (MOOCs), Open Educational Resources (OERs), Open Course Ware (OCW) and open university (*Yousef & Sumner, 2020*). These forms of open learning have the potential to democratize education and provide opportunities for lifelong learning to students around the world (*Tualaulelei & Green, 2022*). While e-learning systems face numerous challenges, their most crucial function is to identify students who are at risk of academic failure and provide timely interventions to support their success. This has resulted in large and complex datasets that are generated by various sources of learner behavior such as how long a student spends on a particular task,

Corresponding author
Shikah Abdullah Albriki Balabied,
salbriki@ub.edu.sa



how many attempts they make, and their performance on quizzes and tests (*Araka et al., 2022*). These datasets can be analyzed to identify patterns and trends, and to gain insights into how learners learn (*Quadir, Chen & Isaias, 2020*). Despite their rapid rise in popularity, OLEs face challenges such as high rates of student failure and inadequate support for learners who are struggling. Many learners who enroll in OLEs do not complete the course, and even those who do complete the course often do not get personal support and feedback (*Elmesalawy et al., 2021*). This form of education also requires learners to be self-motivated and disciplined, as they must manage their own learning and stay on track without the structure and support of a traditional classroom (*Xie, Siau & Nah, 2020*). Thus, the completion rates for OLEs are typically very low and high failure rates in exams that can contribute to a student being considered at-risk (*Shah et al., 2022*). Moreover, at-risk students may struggle with academic performance due to a variety of factors, such as a lack of resources, poor attendance, or a lack of academic support which can interfere with their academic success. One way to address these challenges is to early prediction of at-risk students and deliver the necessary academic and emotional support intervention (*Gupta, Garg & Kumar, 2021*).

There are several approaches that can be used to predict student performance. One common approach involves using machine learning algorithms such as decision trees, random forests, and neural networks that can be used to build predictive models (*Ehsani, Moghadas Nejad & Hajikarimi, 2022*). These models can be trained on large datasets to identify patterns and make predictions about future performance. In order to evaluate the effectiveness of these techniques for early prediction of students' academic failure, several key factors need to be considered. These include the quality and quantity of data available, the accuracy of the predictive models used, and the relevance of the predictors chosen (*Yağcı, 2022*). This study set out to analyze OLEs logs to reveal students who are in danger of academic failure and delivering prompt interventions to assist them in achieving academic success. To achieve this goal, we utilized random forest on the behavioral data sourced from OULAD dataset, which comprises data from a Moodle-like educational system used at Open University (*Kuzilek, Hlosta & Zdrahal, 2017*).

## RELATED WORK

It is important to identify at-risk students early and provide them with the appropriate resources and support to help them succeed academically and socially. Predicting student performance using machine learning techniques has been a topic of interest in recent years (*Altabrawee, Ali & Ajmi, 2019*). *Hlosta et al. (2018)* suggested two activity analysis methods: The Markov chains and General Unary Hypothesis Automaton. The former generates a set of data-explaining rules, while the latter produces probabilities of state transitions that indicate the probability of a student changing their behavior based on previous actions. However, the primary disadvantage of both methods is the complexity of the results obtained.

*Okubo, Shimada & Taniguchi (2017)* introduced a system that utilizes state transition graphs to display learning logs and predict future learning activities and final grades. The system used data from 236 current course students and 209 past course students for

prediction. Markov chains were employed to construct the state transition and experiments were conducted to verify prediction accuracy using ongoing course data. Similarly, *Davis et al. (2016)* analyzed MOOC data from courses with over 100,000 students using Markov chains.

*Haiyang et al. (2018)* developed a probabilistic model using OLEs database and assignment deadlines to forecast student withdrawal risk. Various OLEs types were used to extract features for each student within a specific time frame, achieving an accuracy of 0.84 on the dataset. Dynamic behavioral features and other characteristics were also considered.

*Alshabandar et al. (2018)* utilized assignment deadlines and the VLE database to implement a probabilistic model for dropout prediction. Different types of VLEs were employed to extract features for each student within a specific time interval. Additional features included dynamic behavioral features, demographic features, and various assignment features. The Gaussian Finite Mixture model was used for prediction.

*Dalipi, Imran & Kastrati (2018)* have noted that machine learning is a simple aspect of the process, while the feature selection from the vast amount of data provided by MOOC platforms is the crucial and challenging part. Due to data being presented differently across different platforms, it is often sparse and contains many features. The authors examined two MOOC datasets and segmented them into two categories based on the available features. The first category pertains to student-related features, which primarily comprise the student's behavioral patterns, while the second category pertains to MOOC-related features, which describe the course and its modules. The authors contend that although there is an abundance of data, it still lacks features that can accurately identify not only students at risk of dropping out but also the outcome of their performance.

*Liu & Li (2017)* used the k-means algorithm to cluster active and non-active participants and then applied the Apriori algorithm to uncover association rules for predicting course dropouts with an 80% precision rate. *Chaplot, Rhim & Kim (2015)* analyzed clickstream and sentiment data from Coursera, using sentiment analysis to predict dropout based on sentiment scores. An artificial neural network was used, but the interpretability of the model was limited. The analysis was also constrained by imbalanced data.

*Boyer & Veeramachaneni (2015)* classified students based on their prior enrollment behavior to predict dropouts. Logistic regression was applied to different samples, and the weight distribution was estimated in advance. Various classification algorithms such as decision trees, naïve Bayes, support vector machines, and neural networks have been employed for predicting student performance (*Alturki & Alturki, 2021*). Extracting meaningful features from the dataset, such as click counts, resource usage time, and login frequency, is crucial for accurate learning models.

*Brahim (2022)* created an 86-dimensional feature space using informative features and used multiple machine learning algorithms to classify students as high or low performers, achieving 97.4% accuracy with a random forest classifier.

*Jawad, Shah & Tahir (2022)* utilized the Open University Learning Analytics Dataset (OULAD) to identify areas where students struggled by analyzing assessment scores and engagement metrics. This allowed for targeted interventions and yielded a 96% accuracy rate.

*Kuzilek, Hlosta & Zdrahal (2017)* employed statistical personal and behavioral data in OLEs to predict students' performance using a combined neural network that analyzed both static and sequential data. Different time-series deep neural network models were compared, with GRU and simple RNN performing better than the more complex LSTM model. Over 80% prediction accuracy for at-risk students was achieved by the end of the semester, validated by experiments on the OULAD dataset.

## MATERIALS AND METHODS

This article presented a model for early identification of students at risk applied the random forest algorithm on behavioral data from the Open University Learning Analytics Dataset (OULAD). The following sections will describe more details of this model.

### Dataset

The OULAD (*Kuzilek, Hlosta & Zdrahal, 2017*), a dataset on learning analytics from the Open University, has information on students, courses, and their interactions with the VLE. The dataset includes seven modules named AAA, BBB, CCC, DDD, EEE, FFF, and GGG, and the courses were offered in February and October, represented by B and J respectively. The February semesters are shorter than the October semesters by 20 days. According to the data's original documentation, information on courses CCC, EEE, and GGG is unavailable for 2013 and 2014. Data from 22 modules instructed at the Open University was utilized to create the dataset. It includes demographic information on 32,593 students in addition to their assessment results and interactions with the virtual learning environment (VLE) in the form of clickstreams. The clickstream data is recorded as daily summaries and consists of 10,655,280 entries. Figure 1 shows the database schema of the OULAD dataset, which is used in the study. It is focused on student data rather than course data and includes information on student demographics, activities, and module presentation. The "courses" relation has details on course name (code_module), the year and semester it is offered (code_presentation), and the presentation length in days (length). The "assessments" relation contains information on assessments given in a particular module presentation. All courses have assessments and a final exam, with a combined weight of 100, but in some courses, only exams are weighted (*e.g.*, course GGG). Course GGG also has computer-marked assessments (CMAs) on the same date, and the first assessment is taken after 60 days, whereas in other courses it is taken during the first 30 days. The "VLE" relation includes information on all resources available to students in the VLE, such as pdf files and html pages, and records student interactions with them, including resource identification number (id_site), code_module, code_presentation, activity_type, and the week when the material is scheduled to be used (week_from) and when it is scheduled to be completed (week_to). The "studentInfo" relation provides demographic information on students, while the "studentRegistration" relation records the registration and unregistration dates of a course presentation. The "studentAssessment" relation contains students' assessment results, with a scorerange of 0–100, where a passing score is 40 or higher.

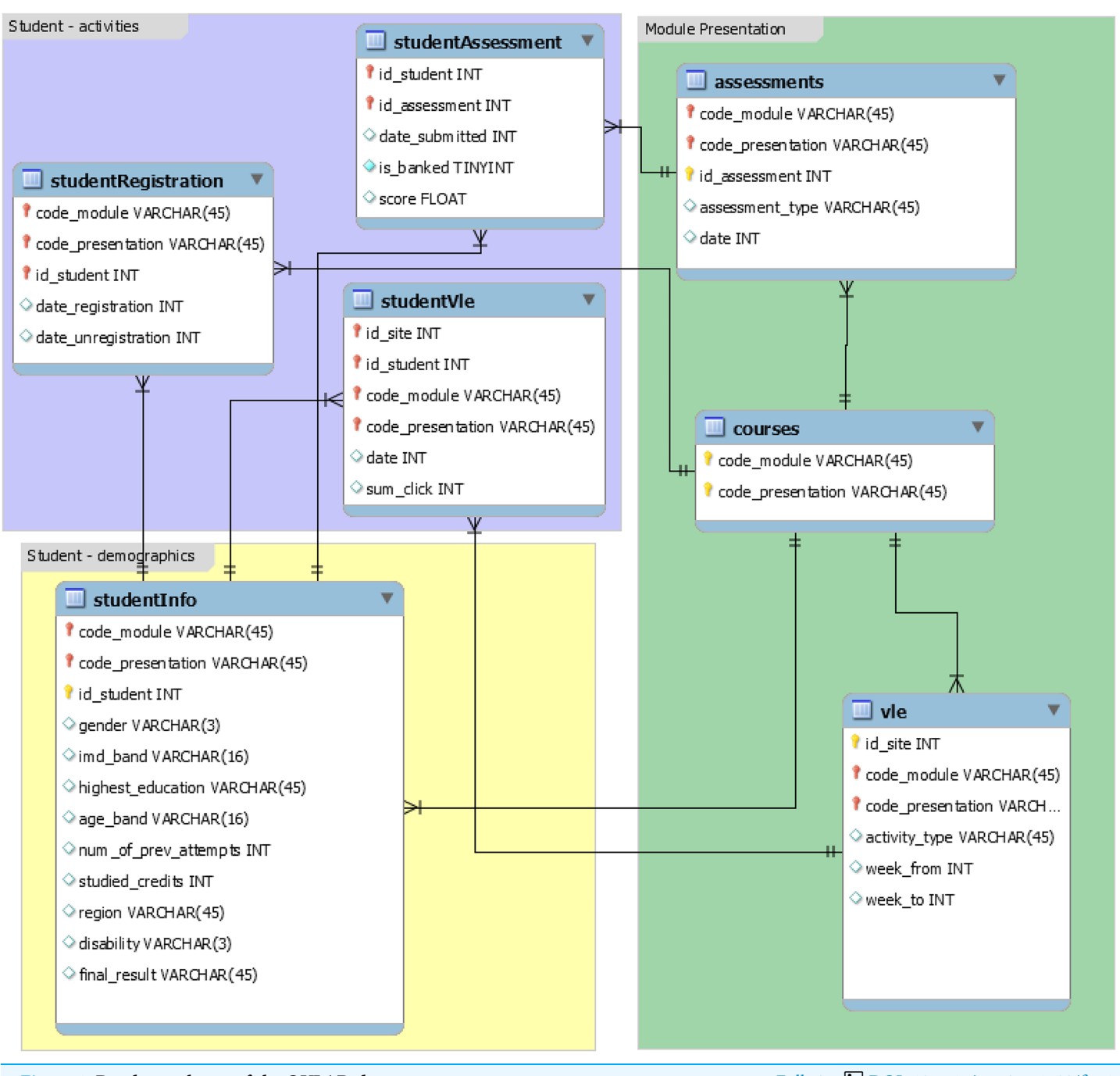

**Figure 1 Database schema of the OULAD dataset.**

## Data preprocessing

To improve the accuracy of model predictions, data preprocessing is necessary, wherein feature engineering methods are utilized. The first method involves filling in missing values in the date column of the assessment data using the mean number of days. The second method involves deleting missing values in the Student Assessment data that are relatively

small in number compared to the total data. In the Student information data, the third method involves filling in missing values in the imd_band column with the mode since it represents a category. Lastly, the fourth method involves deleting the week from and week to columns in the VLE data that have almost 80% missing values.

## Data merging

This stage involves merging various data frames by joining different tables according to the database schema as shown in Fig. 1 and analysis this data.

- Merging the student VLE table with the VLE table; this can provide us with an understanding of how the students interacted with VLE.
- Merging Student Registration table with courses table to investigate the correlation between registrations and the duration of the course.
- Merging Student Info and Student Registration tables.
- Merging Assessments table with Student Assessment table to investigate the correlation between student performance and assessments.
- Merging VLE data with Student Info data.

## Random forest algorithm

The random forest algorithm is an ensemble learning method that combines multiple decision trees to make predictions. It is known for its robustness, accuracy, and ability to handle high-dimensional data with complex relationships consists of two main steps: building the individual decision trees and aggregating their predictions. This study utilized the scikit-learn library's implementation of the random forest algorithm in Python. Then configured the hyperparameters of the random forest, such as the number of trees, maximum depth, and the number of features considered at each split, through a systematic tuning process using techniques like grid search or random search. This allowed us to optimize the model's performance based on the specific characteristics of OULAD dataset.

The accuracy was calculated by comparing the model's predictions with ground truth labels for identifying at-risk students. Specifically, we used metrics such as precision, recall, and F1 score to evaluate the model's performance. These measures consider true positives, false positives, and false negatives, providing a comprehensive assessment of the model's accuracy in identifying at-risk students.

Furthermore, it is crucial to validate the model's performance using a different dataset to assess its generalizability. This study followed a rigorous validation process by splitting the dataset into training and testing subsets. The model was trained on the training set and evaluated on the testing set, which was distinct from the training data. This approach helps to ensure that the model's performance is not solely optimized for the specific dataset employed in the study.

## RESULTS

One of the indicators utilized for identifying student failure is determining if students submit assessments on time or late. There are three types of assessments include Tutor

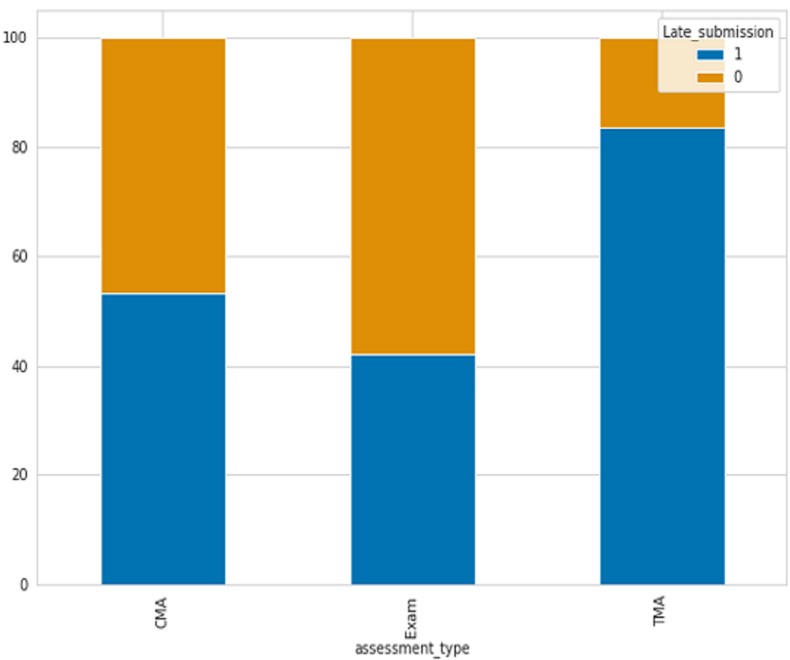

**Figure 2** The percentage of late submissions based on assessments.

Marked Assessment (TMA), Computer Marked Assessment (CMA), and Exams. As shown in Fig. 2 the percentage of late submissions categorized by the Assessment_Type column. The data indicates that the final exam and CMA, which is a computer marked assessment, have the highest number of late submissions, whereas Tutor Marked Assessments have the lowest number of late submissions, where 0 refer to late and 1 not.

After analyzing seven course modules, with four falling under STEM courses (CCC, DDD, EEE, FFF) and the remaining three under social courses (BBB, CCC, and DDD), has revealed that the majority of late submissions were for courses BBB, CCC, and DDD as shown in Fig. 3, where 0 refer to late and 1 not.

STEM courses have a lower percentage of late submissions at 26.07%, while social science courses have a higher percentage of late submissions at 37.07%. According to the table description, scores below 40 are classified as failing while scores above 40 are passing. Figure 4 displays the ratio of passing and failing students, revealing that the failure rate in STEM subjects is higher than in social sciences.

Before the course began, a particular student had multiple interactions, indicating their interest. It is reasonable to believe that if a student is interested in a course, they will engage with its materials before the course starts. The number of clicks made before the course starts could be a significant factor in predicting a student's performance. As a result, we will calculate the number of clicks that each student made both before and after the course started.

After analyzing the correlation between the number of clicks and academic performance as shown in Fig. 5, it is apparent that there is a direct association between the two. The data indicates that students who attained a distinction had a considerably higher number of

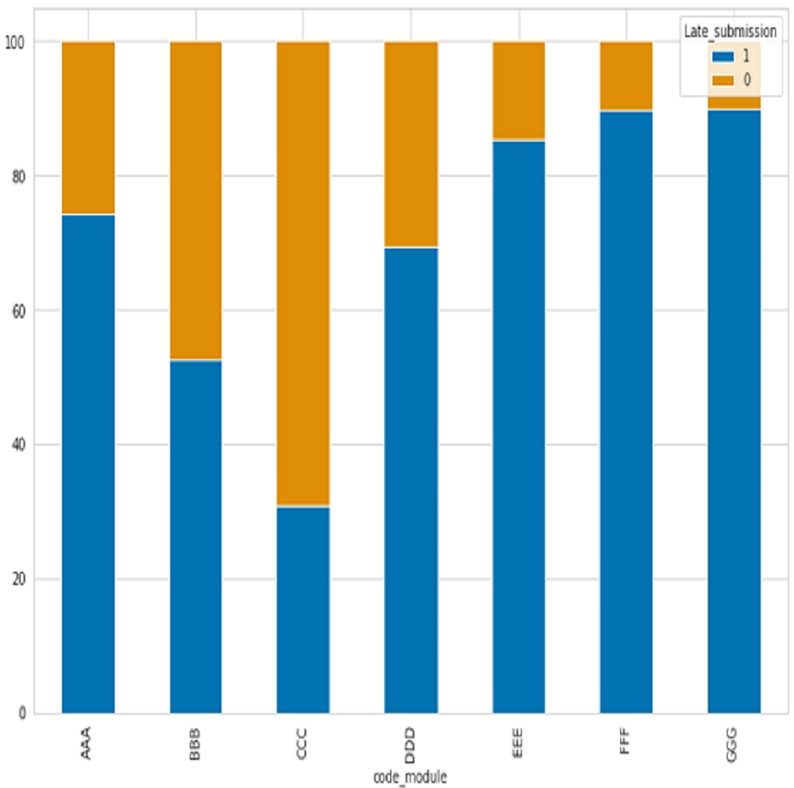

**Figure 3** The percentage of late submissions based on courses.

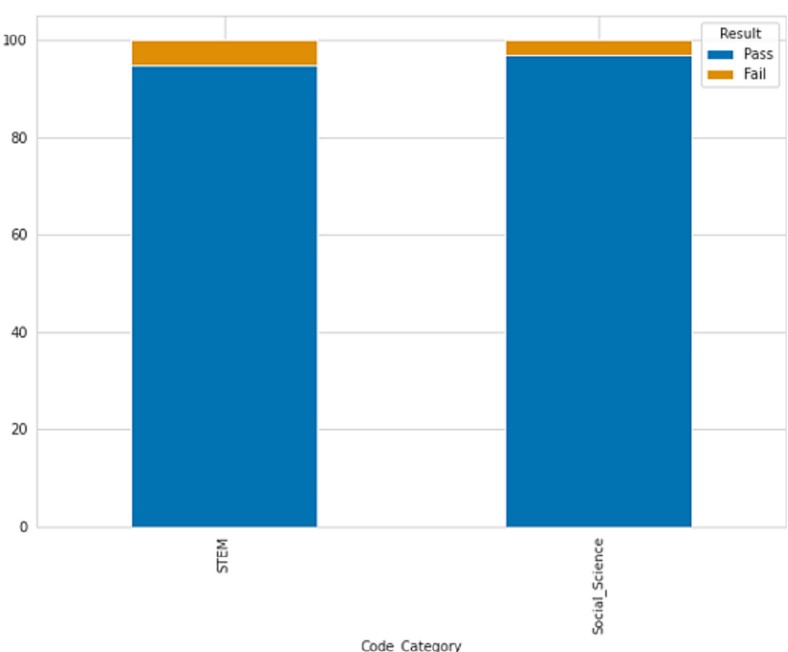

**Figure 4** The percentage of passing and failing students based on courses.

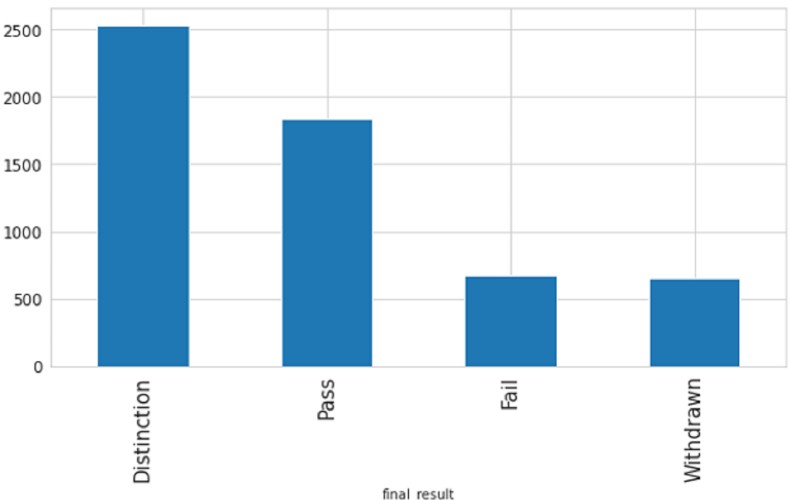

**Figure 5 The pass and fail rates are grouped by the number of clicks.**

clicks than those who received a pass grade, and there was a substantial difference between students who failed or withdrew and those who passed with or without distinction.

After analyzing data on individuals who have earned credits, it was discovered that those with a lower number of credits have a greater probability of passing with distinction or passing in general, presumably because they have to study less. In contrast, those who withdrew have more average credits as compared to another. Figure 6 displays the connection between age and result category. The data indicates that individuals in the age group of 0–35 have a higher rate of failure and withdrawal compared to other age groups.

Based on Fig. 7, it is apparent that individuals with no formal education or education lower than level education have higher rates of failure and withdrawal, while those with a post-graduate degree have the lowest failure rate.

The random forest classifier is a machine learning technique that can perform both regression and classification tasks. It creates multiple decision trees that are trained on different subsets of the same training set to enhance the classification accuracy and address the issue of over fitting (*Mahboob, Irfan & Karamat, 2016*). The random forest algorithm creates K number of trees by randomly selecting attributes without pruning. In contrast to the decision tree method, which tests the test data on a single constructed tree, random forest tests the test data on all the built trees and assigns the most common output to that instance (*Mishra, Kumar & Gupta, 2014*). Typically, a forest with more trees is more robust. The random forest classifier follows this principle, producing the most accurate results when it has a higher number of trees. Additionally, random forest can handle missing values and categorical data, and it avoids over fitting. To measure the purity and impurity of attributes, random forest uses the Gini index.

The Gini index is a method for selecting an attribute that measures the impurity of an attribute in relation to a dataset's classes. Like entropy, the Gini index reaches its maximum value when all classes in the dataset have equal probability. The formula for the Gini index is defined as (*Pal, 2005*):

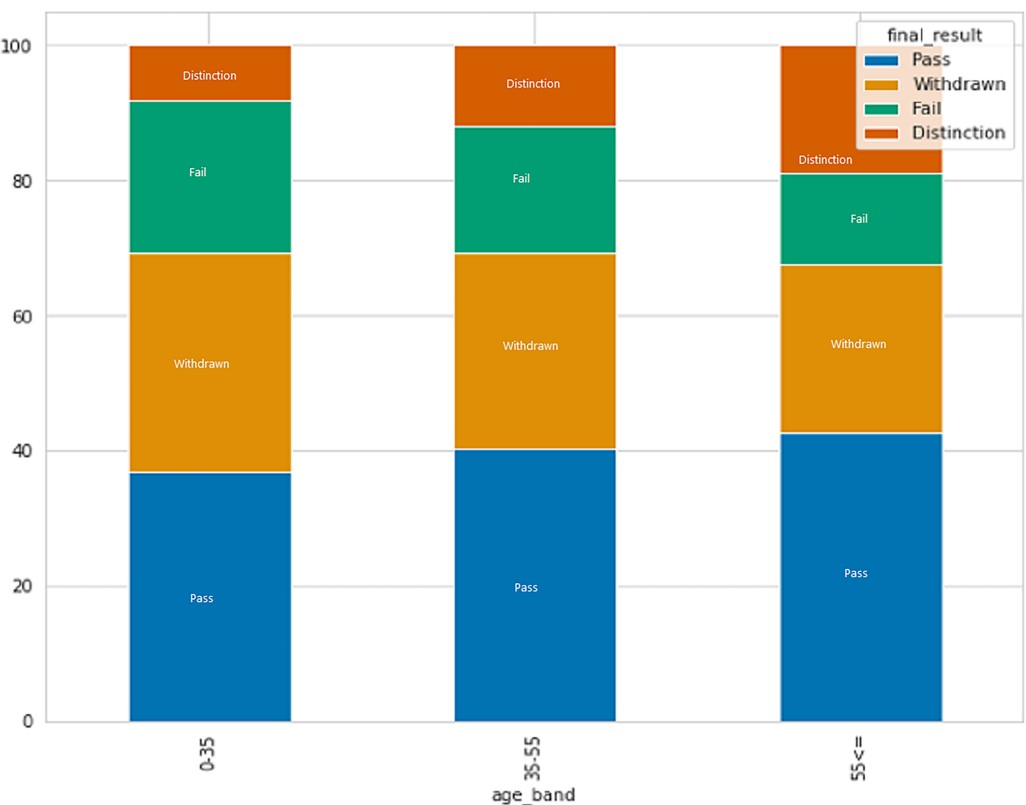

**Figure 6 The relation between age and result category.**

$$\text{Gini(t)} = 1 - \sum_{i=0}^{c-1} \left[ p(i|t) \right]^2$$

In this article, the analysis data described in Section Material and Method was utilized as the input for a random forest classifier to predict student failure at an early stage illustrated in Fig. 8. Random forest is a classification method that consists of many decision trees. It employs bagging and feature randomness while constructing each tree to create an independent forest of trees. By combining the predictions of each tree, the resulting committee prediction is typically more precise than that of any individual tree.

The accuracy for early prediction of student failure using a random forest classifier based on the OULAD dataset. The dataset was divided into training and testing sets, with 80% of the data allocated for training and 20% for testing. The objective here is to find features which impact the decision whether student will fail or not, and as such, the majority of the data will be utilized for training purposes only. This will involve initializing a random forest model, fitting it with the training data, and subsequently using the fitted model to predict the outcomes of the test data. The results of this process are presented in Table 1.

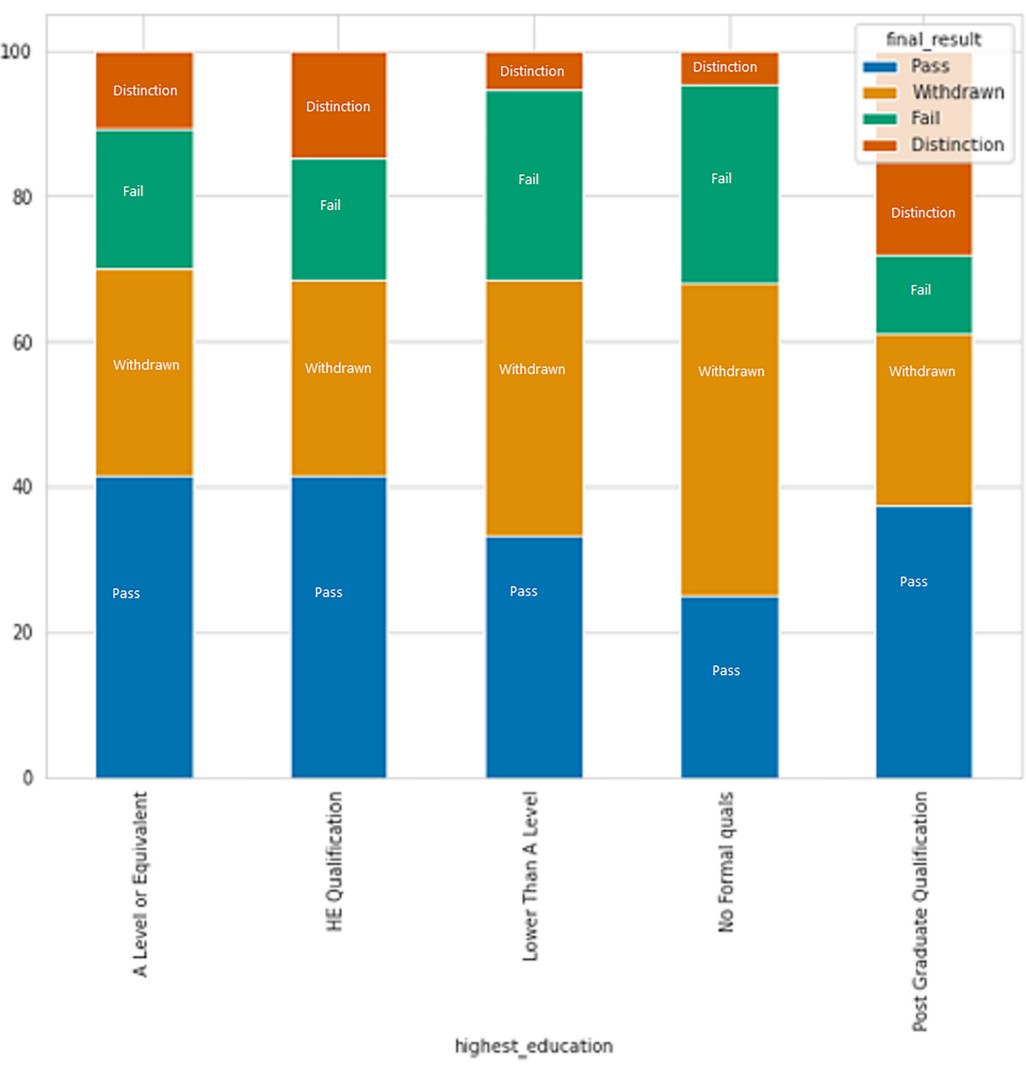

**Figure 7 The relation between formal education and result category.**

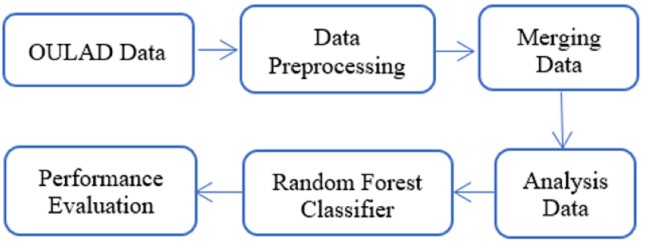

**Figure 8 Model architecture.**

## DISCUSSION

Students who are at risk can be identified by several factors, including low participation or attendance in classes, extracurricular activities, or online resources, poor academic performance, missed deadlines or assignments, a lack of communication or feedback, and

**Table 1 The result of prediction model.**

|  | Precision | Recall | F1-score | Support |
|---|---|---|---|---|
| 0 | 0.85 | 0.95 | 0.90 | 3,077 |
| 1 | 0.95 | 0.85 | 0.90 | 3,442 |
| Accuracy |  |  | 0.90 | 6,519 |
| Macro avg. | 0.90 | 0.90 | 0.90 | 6,519 |
| Weighted avg. | 0.90 | 0.90 | 0.90 | 6,519 |

symptoms of stress, anxiety, depression, or other mental health conditions. Additionally, lack of academic or career objectives or plans, a lack of social or academic support networks. It can be seen from the data in Table 1 that the proposed model achieving classification accuracy of 90% using a Random Forests algorithm to predict students at risk suggests that the model is performing relatively well in identifying students who are likely to be at risk of academic failure or dropping out. The proposed approach classifies them as a group due to the binary nature of the problem. This means that the model treats all students who are at risk of failure as a single group, rather than identifying each student individually. This approach can help maintain the privacy of individual students while still providing valuable insights to teachers and administrators.

The purpose of this study is to predict student performance early on in order to prevent dropouts and improve the online learning experience through timely interventions and support. Early performance is evaluated based on two criteria: academic performance, which is measured by student scores, and academic engagement, which is measured by the frequency of clicks on course content. The data shows that the number of clicks for students who achieved a Distinction was significantly higher than those who received a Pass grade. There was a notable difference between students who failed or withdrew and those who passed with or without distinction. Furthermore, determining whether students submit assessments on time or late is critical in detecting early student failure. The analysis of data also reveals that individuals with no formal education or education below a certain level have higher rates of failure and withdrawal, whereas those with a post-graduate degree have the lowest failure rate. All of these indicators help to identify early predicting student failure.

By accurately predicting the likelihood of success or failure for the entire class, teachers can take appropriate measures to enhance the overall performance of the students (*Raj & Renumol, 2022*). For example, educators may provide additional feedback and resources to students who are at risk of failure or adjust the curriculum to better meet their needs (*Nachouki & Abou Naaj, 2022*). Taken together, these results suggest that the proposed random forest model can be a valuable tool in improving student outcomes and promoting academic success. The findings of this study have a number of important implications for future practice as it can help stakeholders understand and address the factors leading to academic failure. Deploy the model in a real-world setting, such as a school or university, to start predicting at-risk students early and intervene accordingly.

### Limitations

The model uses random forest to analyze these factors and identify patterns that are associated with successful or unsuccessful outcomes. It then applies these patterns to new data to make predictions about which students are at risk of failing the course. However, it's important to note that the accuracy metric alone may not be enough to fully evaluate the model's performance (*Hu, 2022*). For instance, it's possible that the model correctly classifies a high percentage of students who are not at risk but is not as accurate in identifying those who are truly at risk (*Tamada, Giusti & Netto, 2022*). In such cases, other metrics such as precision, recall, or F1 score may be more informative in evaluating the model's performance.

Additionally, it's important to consider the context in which the model was developed and evaluated. For example, the model may have been trained on a particular dataset *i.e.*, OULAD, and its accuracy may not generalize well to other contexts or populations. Therefore, it may be necessary to evaluate the model's performance on new datasets and regularly update the model with new data to ensure it remains relevant and accurate before making decisions based on the model's predictions.

## CONCLUSIONS

Machine learning techniques can be effective in early prediction of at-risk students by analyzing large datasets to identify patterns and relationships among various factors that contribute to student success or failure. The random forest algorithm has been applied on behavioral data from the Open University Learning Analytics Dataset (OULAD) to predict student performance. The dataset was divided into training and testing sets, with 80% of the data used for training and 20% for testing. The objective is to identify the key features that influence student failure, and the majority of the data is utilized for training purposes only. This involves initializing a random forest model, fitting it with the training data, and using the fitted model to predict the outcomes of the test data. The results show an accuracy rate of 90%. The results of this study indicate that the random forest based model provides a powerful tool for identifying students who may be at risk of failure and guiding them towards success. Considerably more work will need to be done to improve the accuracy and efficiency of predicting student performance using machine learning. This includes incorporating novel techniques like active learning, transfer learning, and incorporating student emotions and engagement. Thus, we suggest that to have more data where we can create three separate datasets for training, validation and testing to enhance the models' accuracy and performance. While this study focused specifically on the random forest classifier, future studies should conduct a comparative analysis with other widely used methods such as XGBoost, AdaBoost, neural networks, and others would provide valuable insights into the performance and effectiveness of different algorithms in this context.

### Funding
This work was supported by the Deanship of Scientific Research, University of Bisha through the general research project under grant number (UB-GRP-27-1444). The funders had no role in study design, data collection and analysis, decision to publish, or preparation of the manuscript.

### Grant Disclosures
The following grant information was disclosed by the authors:
Deanship of Scientific Research, University of Bisha: UB-GRP-27-1444.

### Competing Interests
The authors declare that they have no competing interests.

### Author Contributions
- Shikah Abdullah Albriki Balabied conceived and designed the experiments, performed the computation work, prepared figures and/or tables, authored or reviewed drafts of the article, and approved the final draft.
- Hala F. Eid performed the experiments, analyzed the data, prepared figures and/or tables, and approved the final draft.

### Data Availability
The data is available at Open University Learning Analytics: https://analyse.kmi.open.ac.uk/open_dataset.

The raw measurements are available in the Supplemental File.

### Supplemental Information
Supplemental information for this article can be found online at http://dx.doi.org/10.7717/peerj-cs.1708#supplemental-information.

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
