# Peer review of "Utilizing random forest algorithm for early detection of academic underperformance in open learning environments"

_PeerJ Computer Science, doi:10.7717/peerj-cs.1708_

## Round 0.1 · original submission · Major Revisions

The paper needs an extensive revision by the author, and the comments provided by the reviewers must be addressed.

Reviewer 1 ·

Basic reporting

In the related work, the authors state “It's important to identify …” Do not use abbreviations. Instead, use “It is important to identify …”

Some typos and grammar errors should be revised for better readability.

Experimental design

The Random Forest classifier model is used to analyze anonymized large datasets from Open University Learning Analytics to identify patterns and relationships among various factors that contribute to student success or failure. The experimental results suggest that the algorithm achieved 90% accuracy in identifying at-risk students and providing them with the necessary support to succeed.

However, do the authors try comparing the Random Forest with other state-of-the-art methods that are widely used in academia and industry, like XGBoost, AdaBoost, Neural Networks, etc.?

Validity of the findings

valid

Additional comments

More details about Random Forest should be elaborated in Section Materials & Methods.

In Figure 2, the late_submission is kind of confusing for readers. Some people may think late_submission = 1 means the student submitted late.

Reviewer 2 ·

Basic reporting

In this article, the authors proposed a model that could identify students' academic risk in an educational environment. The paper lacks an in-depth discussion of the problem formation and the proposed solution. The manuscript lacks in different aspects; some major concerns are as follows.

Experimental design

The introduction to Open Learning Environments (OLEs) and their scalability in the abstract might be more succinctly and narrowly targeted to better convey the specific study challenge. It's critical to establish the study topic or problem statement up front.

Although the Random Forest classifier model is discussed in the abstract, there are no specifics on the characteristics and data that were utilized to create the model. It would be more thorough of the abstract to give a brief description of the variables and data sources used.


The assertion that at-risk students may be identified with a 90 percent accuracy is encouraging, but it would be helpful to know how this accuracy was calculated and whether the model's performance was validated using a different dataset. Furthermore, a discussion of the model's potential generalizability outside of the dataset employed would deepen the conclusions.

Validity of the findings

The assertion that at-risk students may be identified with a 90 percent accuracy is encouraging, but it would be helpful to know how this accuracy was calculated and whether the model's performance was validated using a different dataset. Furthermore, a discussion of the model's potential generalizability outside of the dataset employed would deepen the conclusions.
Potential study limitations should be acknowledged in a thorough abstract. A thorough knowledge of the scope of the research would show that any constraints, such as biases in the dataset or the model's sensitivity to particular factors, were mentioned.

Additional comments

Although it is briefly mentioned in the abstract that early treatments were given to at-risk pupils, it should go into more detail about the application of the research. What implications might these findings have for enhancing the learning environment in OLEs? What difficulties would there be in putting these solutions into practice on a large scale?
Readers can gain an understanding of the significance of the research beyond the results by reading a phrase or two on possible future research directions or the study's overall impact in the abstract.
The overall language of the paper needs professional improvements. for example, a large number of grammatical errors can be seen throughout the manuscript.
the abstract is too general and needs serious attentions.

---

## Round 0.2 · accepted · Accept

The paper is of a very good standard.

Reviewer 1 ·

Basic reporting

The authors addressed all my concerns. I agree to accept this paper for publication.

Experimental design

The authors addressed all my concerns. I agree to accept this paper for publication.

Validity of the findings

The authors addressed all my concerns. I agree to accept this paper for publication.

Reviewer 2 ·

Basic reporting

The paper seems to be revised well,

Experimental design

Satisfied

Validity of the findings

Satisfied with the revised version of the paper and has no more comments

Additional comments

No additional comments